# Simultaneous Use of Stimulatory Agents to Enhance the Production and Hypoglycaemic Activity of Polysaccharides from *Inonotus obliquus* by Submerged Fermentation

**DOI:** 10.3390/molecules24234400

**Published:** 2019-12-02

**Authors:** Mengya Wang, Zhezhen Zhao, Xia Zhou, Jinrong Hu, Jiao Xue, Xiao Liu, Jingsheng Zhang, Ping Liu, Shisheng Tong

**Affiliations:** 1College of Food Science and Nutritional Engineering, China Agricultural University, Beijing 100083, China; 18336778891@163.com (M.W.); zhaozz9789@163.com (Z.Z.); hujr@cau.edu.cn (J.H.); xj1154009143@163.com (J.X.); liuxiao199508@163.com (X.L.); huaxue@cau.edu.cn (J.Z.); 2Bio-Pharmaceutical College, Beijing City University, Beijing 100094, China

**Keywords:** *Inonotus obliquus*, polysaccharide, submerged fermentation, stimulatory agents, α-glucosidase, monosaccharide composition

## Abstract

This study aimed to determine the effect of applying stimulatory agents to liquid cultured *Inonotus obliquus* on the simultaneous accumulation of exo-polysaccharides (EPS) and their monosaccharide composition. Different stimulatory agents (VB_6_, VB_1_, betulin and birch extract) were investigated for their effects on active exo-polysaccharides by submerged fermentation of *I. obliquus*. The mycelial biomass, reducing sugar content, EPS yield and α-glucosidase inhibition rate were determined, and the EPS obtained was analyzed for monosaccharide composition. The results showed that the addition of all the four stimulatory agents could significantly increase the inhibitory activity against α-glucosidase of EPS than the control, whereas EPS from 4 μg/mL VB_1_-containing medium had the best effect with an estimated IC_50_ value 24.34 μg/mL. Among the four stimulatory agents, VB_6_ gave maximum production of mycelial biomass and EPS at the concentration of 4 μg/mL with a increase of 50.79% and 114.46%, respectively. In addition, betulin had a significant effect on increasing the EPS yield and activity, and birch extract had a significantly stimulatory effect on the mycelial growth and the polysaccharides activity, only slightly worse than VB_6_ and VB_1_. Moreover, the addition of different stimulatory agents changed the monosaccharide composition of polysaccharides, which had a correlation with polysaccharide activity.

## 1. Introduction

*Inonotus obliquus (Fr.) Pilát* is an edible fungus with medicinal properties. It has been used to treat gastrointestinal cancer, cardiovascular disease and diabetes in the folk since the 16th century [1]. The chemical substances isolated from *I. obliquus* to date, such as polysaccharides, triterpenoids, steroids exhibit multiple biological activities [2,3]. Based on the presence of several biologically active compounds, *I. obliquus* has many important functions, including hypoglycemic, antioxidant, immune-stimulating and antitumor effects [4,5,6,7,8]. Polysaccharides are important secondary metabolites from the edible and medicinal mushroom *I. obliquus*. The structure and activity of polysaccharides produced in the submerged fermentation of *I. obliquus* have been widely studied. Polysaccharides from *I. obliquus* can reduce postprandial blood glucose and achieve the treatment of diabetes by inhibiting intestinal α-glucosidase and reducing the absorption of carbohydrates in food [9], which has promising prospects for the development of natural non-toxic hypoglycemic drugs.

Recently, liquid submerged fermentation has been widely used in the culture of *I. obliquus* due to its shorter cycle, higher yield and lower price. Both the rarity of *I. obliquus* fruit bodies and the low efficiency of the current submerged fermentation method lead to a low yield of polysaccharides. In order to improve the yield and activity of polysaccharides in submerged fermentation, the fermentation medium components have been extensively studied, such as carbon sources, nitrogen sources, inorganic salts and pH [10,11,12]. However, these traditional methods do not lead to any significant improvement in polysaccharide activity. In recent years, the addition of exogenous substances such as promoters, growth factors, and Chinese medicinal materials to submerged fermentations has attracted more attention. It has been reported that various stimulatory agents have an important influence on the growth state, yield and active components of fungal mycelia [13,14]. 

Vitamins often contribute to biocatalyst-enzyme activities in the form of coenzymes or prosthetic groups, promoting the formation and transfer of methyl groups, advancing carbohydrate metabolism in tissues surrounding cells, and participating in important life activities such as protein and fat metabolism [15]. In liquid submerged fermentation, VB_1_, VB_6_, and VB_12_, which are important stimulatory agents in the liquid fermentation medium of *Inonotus obliquus*, are often added as growth factors to promote mycelium growth and metabolite generation [15,16]. Betulin, featuring anti-tumor, anti-inflammatory, and anti-allergy effects, was extracted from birch bark. Studies have found that the addition of betulin as a stimulating factor to the fermentation medium of *Inonotus obliquus* can significantly increase the levels of metabolites such as triterpenoids and steroids [17]. Some studies have found that adding birch extract to the culture medium by water extraction can significantly promote the mycelia growth of *Inonotus obliquus* and the steroid content [18,19,20]. The nutrients of birch extract, which mainly contains fatty acids, polyphenols, triterpenoids, betulin and betulinic acid were studied by HPLC and GC-MS [19,21]. Moreover, it can be used as an inducer to induce the production of active polysaccharides, which provides a good direction for future research. 

However, little attention has been paid to the effects of culture medium on the monosaccharide composition of EPS, or to the relationship between monosaccharide composition and biological activity. The goal of this work was to demonstrate the feasibility of an approach for simultaneously enhancing the production and bioactivity of polysaccharides from *Inonotus obliquus* in submerged cultures. We also analyzed the monosaccharide composition of the polysaccharides from *Inonotus obliquus* obtained under the conditions of adding different stimulatory agents, in an attempt to explore the relationship between the composition of monosaccharides and the activity of polysaccharides, in order to prove that the addition of stimulatory agents in the medium can be used as a promotion of a new way to produce high activity polysaccharides by submerged fermentation of *Inonotus obliquus*. 

## 2. Results

### 2.1. Effect of VB_6_ on Bioactive Exopolysaccharides by Submerged Fermentation of Inonotus Obliquus

Figure 1 shows the effects of different concentrations of VB_6_ on the growth, polysaccharide content and activity of *Inonotus obliquus.* The results show that the addition of VB_6_ had a stimulatory effect on mycelial biomass, which reached the highest value at 4.75 g/L, a significant increase of 50.8% compared with the control. Besides, the trend of reducing sugars in the fermentation broth was contrary to that of biomass, and the reducing sugar reached the lowest value at 4 μg/mL of VB_6_. This result indicated that the addition of VB_6_ has a stimulatory effect on mycelia growth and reducing sugar consumption, whereas the best stimulation effect was exhibited at the concentration of 4 µg/mL. On the other hand, polysaccharide production significantly increased with the addition of VB_6_ compared to the control, and the addition of 4 μg/mL VB_6_ increased polysaccharide production with maximum stimulation, reaching 481.71 mg/g which represents a significant enhancement of 114.5%. However, when the VB_6_ concentration exceeded 4 μg/mL, the polysaccharide production obviously decreased, and was even lower than control. This showed that VB_6_ was able to increase the polysaccharide production in a certain extent, but an excessively high concentration of VB_6_ inhibited the synthesis of polysaccharides from *I. obliquus.* Combined with the trend of mycelial biomass, a significant positive correlation could be found between the EPS yield and the mycelial growth (*p* < 0.05).

With the increase of VB_6_ concentration, the bioactivity of the *Inonotus obliquus* polysaccharides also increased. Moreover, the best activity was exhibited at the concentration of 4 µg/mL and the IC_50_ reached the lowest value of 48.88 μg/mL and significantly (*p* < 0.05) decreased by 86.7%, compared with the control. However, the activity was decreased when the concentration of VB_6_ was higher than 4 μg/mL. According to the data (Figure 1), it might be concluded that the change trend of the α-glucosidase inhibitory activity is consistent with those of biomass and EPS production, indicating that the biomass and EPS production have a certain positive correlation with the activity of the polysaccharides under VB_6_ stimulation. 

Figure 2 depicts the chromatograms of seven standard monosaccharides (glucose, rhamnose, galactose, arabinose, xylose, mannose, and fucose). The results for monosaccharide composition of *Inonotus obliquus* polysaccharides with different concentrations of VB_6_ are presented in Table 1. The molar ratio of rhamnose and fucose tended to firstly increase and then decrease in association with an increased concentration of VB_6__._ When the concentration of VB_6_ was 4 μg/mL, the polysaccharide had the highest activity, at the same time, the mole percentage of rhamnose and fucose also reached the highest. Pearson correlation analysis showed that under the stimulation of VB_6_ the α-glucosidase inhibitory activity of polysaccharides from *Inonotus obliquus* had a significantly positive correlation with the percentage of rhamnose in the monosaccharide composition (*p* < 0.05). This indicated that there is a certain relationship between monosaccharide composition and polysaccharide activity. Kim [22] found that rhamnose has a greater impact on the biological activity of polysaccharides from *Inonotus obliquus.* Furthermore, the change of monosaccharide composition of fermented polysaccharide may be one of the reasons for the change of polysaccharide activity [23].

### 2.2. Effect of VB_1_ on Bioactive Polysaccharides by Submerged Fermentation of Inonotus Obliquus

As shown in Figure 3, as the concentration of VB_1_ increased, the mycelial biomass also increased. The maximum value reached 3.5 g/L at a concentration of 4 μg/mL, and the promotion rate was 11.1%. However, the mycelial biomass was obviously decreased when the concentration of VB_1_ was higher than 4 μg/mL. This indicated that a certain concentration of VB_1_ could promote the mycelial growth. Similarly, the trend of reducing sugar in fermentation broth was contrary to that of biomass, and the reducing sugar reached the lowest (7.93 mg/mL) when the concentration of VB_1_ was 4 μg/mL. However, when the concentration of VB_1_ was higher than 4 μg/mL, the biomass was decreased and the reducing sugar level was increased. This demonstrated once again that there was a negative correlation between biomass and reducing sugar content.

In addition, the EPS yield tended to increase in association with the raised concertration of VB_1_ (Figure 3). The highest EPS yield (385.28 mg/g), representing with a significant increase of 71.5% compared with the control was observed when the concentration of VB_1_ was set at 6 μg/mL. This indicated that VB_1_ could stimulate the EPS yield from *I. obliquus*, even at a concentration of 5, 6 μg/mL, whereas the biomass was decreased. Moreover, it could be seen that the addition of VB_1_ could improve the activity of the polysaccharides. Furthermore, the IC_50_ decreased with the increase of VB_1_ concentration, which reached the lowest (24.34 μg/mL) at 4 μg/mL, with a significant decrease of 93.4% compared with the control. It indicated that a certain concentration of VB_1_ stimulated the metabolic pathway of active polysaccharide synthesis of *Inonotus obliquus*, and produced active polysaccharide but the polysaccharide content did not reach the highest yield. It might be concluded that there was no correlation between α-glucosidase inhibitory activity and EPS yield under the stimulation of VB_1._

Table 2 shows that as the concentration of VB_1_ increased, the activity of polysaccharides gradually increased and the mole ratio of glucose and fucose in the monosaccharide composition also gradually increased. At the concentrations of 4 and 5 μg/mL, the α-glucosidase inhibition rate of polysaccharide reached the highest and the mole percentage of glucose and fucose was at a high level. When the concentration of VB_1_ was 6 μg/mL, the polysaccharide activity decreased, at the same time, the mole ratio of fucose also decreased, while that of glucose increased slightly, indicating that VB_1_ promotes the synthesis of polysaccharides containing a large amount of glucose. Therefore, it might be concluded that the fucose in the monosaccharide composition of the polysaccharides of *Inonotus obliquus* may be positively correlated with the α-glucosidase inhibitory activity of polysaccharides when VB_1_ was added.

### 2.3. Effect of Betulin on Bioactive Polysaccharides by Submerged Fermentation of Inonotus Obliquus

Figure 4 shows the effects of betulin addition to the fermentation medium on growth and active polysaccharides of *I. obliquus*. As can be seen, the mycelial biomass gradually increased with the addition of betulin, indicating that a certain concentration of betulin could promote the mycelial growth. On the other hand, the EPS yield tended to firstly increase and then decrease in association with increased concentrations of betulin. The maximum value (437.5 mg/g) was exhibited at the concentration of 4 μg/mL, with a significant increase of 91% compared with the control. The reason may be that a certain amount of terpenoids increases the fluidity of the cell membrane, causing more polysaccharides to flow out of the cell, resulting in a significant increase in the accumulation of polysaccharide [16]. 

In addition, all tested polysaccharide of the experimental group added with betulin exhibited lower IC_50_ values than the control. Moreover, when 2 μg/mL betulin was added into the medium, the α-glucosidase inhibition rate reached the highest, whereas the IC_50_ value was 82.97 μg/mL. It indicated that low concentration of betulin had a certain promoting effect on the secondary metabolites of *Inonotus obliquus*. On the other hand, it was also found that the effect of betulin on the relationship between mycelium of *Inonotus obliquus*, content of polysaccharides and the effect of α-glucosidase inhibitory activity are not directly related. Based on the present results, it may be concluded that betulin had a promoting effect on the activity of the polysaccharides of *I. obliquus* and the best effect was observed at the concentration of 2 µg/mL.

It can be seen from Table 3 that when adding betulin, a certain proportion of arabinose appears in the monosaccharide composition of the polysaccharides of *I. obliquus*. Moreover, the trend of arabinose was not consistent with that of polysaccharide activity, therefore, it was speculated that it might be involved in the formation of other polysaccharides without α-glucosidase inhibitory activity. As the concentration of betulin increased, the mole ratio of mannose showed a similar trend with polysaccharide activity. The highest activity of EPS was observed at the concentration of 2 μg/mL, whereas the mole ratio of mannose reached the lowest. Besides, as the polysaccharide activity decreased, the mole percentage of mannose increased. Therefore, it can be concluded that the activity of polysaccharide under the stimulation of betulin may be related to the ratio of mannose in its monosaccharide composition.

### 2.4. Effect of Birch Extract on Bioactive Polysaccharides by Submerged Fermentation of Inonotus Obliquus

As shown in Figure 5, the biomass of mycelium gradually increases as the concentration of birch extract increases, reached the highest value (4.65 g/L) at 60% with a significant increase of 47.6% compared to the control, whereas the reducing sugar level reached the lowest value, indicating that the addition of birch extract had a stimulatory effect on mycelial biomass. With the increase of the concentration of birch extract, the polysaccharide content decreased gradually, and it was lower than that of the control group. It may be that the excess nutrients and natural products in the birch extract were mainly used to produce mycelium structure. Birch extract contains a large amount of fatty acids and the addition of fatty acids increases the unsaturated fatty acids in the cell membrane, which makes the cells more susceptible to free radical attack and leads to lipid peroxidation of the cell membrane, which may affect the activity of enzymes involved in polysaccharide synthesis. Therefore, the synthesis ability of *Inonotus obliquus* polysaccharides induced by birch extract decreased, and the polysaccharide content also decreased significantly.

It can be seen from the figure that the α-glucosidase inhibitory activity of the extracellular polysaccharide increases first and then decreases. When the addition of birch extract was 40%, the IC_50_ reached the lowest at 24.92 μg/mL and accounted for a significant decrease of 93.2% compared with the control, indicating that the nutrients in the birch extract could promote the synthesis of high activity EPS by submerged fermentation of *Inonotus obliquus*. The results showed that birch extract could promote the growth of mycelium of *Inonotus obliquus*. Moreover, the EPS yield gradually decreased with the increase of birch extract, which might be due to the fact that the nutrients in the birch extract mainly had a stimulatory effect on the α-glucosidase inhibitory activity of the EPS and mycelial biomass. 

As can be seen in Table 4, the addition of birch extract significantly changed the monosaccharide composition of polysaccharides of *I. obliquus*. When the concentration of birch extract is 20%, 40%, 60%, the polysaccharide containing galactose and xylose. With the increase of birch extract, the mole ratio of xylose increased first and then decreased, while the mole ratio of arabinose decreased first and then increased. When birch extract added was 40%, the α-glucosidase inhibitory activity of the polysaccharides reached the peak, whereas the percentage of arabinose was the lowest and the percentage of xylose was also at a high level. The results indicated that under the stimulation of birch extract the activity of polysaccharides might have a negative correlation with arabinose and a positive correlation with xylose in monosaccharide composition. In addition, it can be found that when adding birch extract, there was no fucose in the monosaccharide composition, and six purified polysaccharides are isolated in our previous studies, wherein the monosaccharide composition of polysaccharide HIOP1-S and HIOP2 and the addition of birch monosaccharide composition are close, but the ratio is different, indicating that the addition of birch extract affects the proportion of monosaccharides of *I. obliquus* polysaccharides [9,23].

## 3. Discussion

In this study, the results showed that the addition of all the four stimulatory agents could significantly increase the inhibitory activity against α-glucosidase of EPS than the control, whereas VB_1_ was the most effective. In addition, VB_6_ gave maximum production of mycelial biomass and EPS. Birch extract showed a significantly stimulatory effect on the mycelial growth and the polysaccharides activity, only slightly worse than VB_6_ and VB_1_, respectively. This indicated that the presence of VB_6_, VB_1_ and betulin in birch extract was necessary for the growth of mycelium and the production of active polysaccharides of *I. obliquus.* On the other hand, based on the results of monosaccharide composition analysis, it might be concluded that the addition of stimulatory agents affected the monosaccharide composition of polysaccharides from *Inonotus obliquus*. Among the four stimulatory agents, only betulin and birch extract could promote the synthesis of polysaccharides containing arabinose, while VB_1_ and VB_6_ cannot. It might be thought that the addition of VB_1_ and VB_6_ had a similar effect on the monosaccharide composition, since they both are vitamins. In the contrary, birch extract resulted in the appearance of arabinose in monosaccharide composition, which might be due to the fact that birch extract contains betulin or contains both VB_1_ and VB_6_, which have a synergistic effect. Furthermore, the monosaccharide composition of the polysaccharide obtained under the stimulation of birch extract was quite different from the other three, whereas there was no fucose, but galactose and xylose. It indicated birch extract contained other nutrients, which made the synthesis of polysaccharides containing galactose and xylose and needed further research. Based on the previous reports, the α-glucosidase inhibitory activity of polysaccharides might be attributed to many factors, including other molecular weight and structure [9,24].

The synthesis of polysaccharides is a complex process [25,26]. The initial carbon source in the culture medium provided sufficient substrate for the synthesis of the polysaccharide by *Inonotus obliquus*. Under the catalysis of enzymes, glucose acts as a sugar donor, while multiple UDP-monosaccharides form polysaccharides under the formation of glycosidic bonds. The addition of stimulating factors promotes the catalysis of different enzymes to affect the composition of monosaccharides, which affect the activity of polysaccharides. However, genetic engineering would have to be relied on to fully understand the synthetic approach, which points out a new direction for future research works. Further studies can be used to study the metabolic process of *Inonotus obliquus* by transcriptome analysis, analyze the function of differentially expressed genes of *Inonotus obliquus* under stimulating factors and the metabolic pathways involved, aiming to clarify the stimulant regulation of the metabolism of *Inonotus obliquus*. The mechanism may lay a theoretical foundation for the realization of metabolic regulation at the gene level.

## 4. Materials and Methods

### 4.1. Materials and Regents

The fruiting bodies of *Inonotus obliquus* were purchased from the Sichuan Institute of Edible Fungi. Betulin and α-glucosidase were purchased from Sigma-Aldrich Co. (St. Louis, MO, USA). 4-Nitrophenyl-α-d-glucopyranoside (PNPG) and phosphate buffer solution (PBS) were purchased from Solarbio Technology Co., LTD (Beijing, China). Birch bark was purchased from the Birch Forest Factory of Changbai Mountain, in Jilin Province, China. All other chemicals and solvents were of analytical grade.

### 4.2. Submerged Fermentation

After drying in an oven at 60 °C, the birch bark was broken with a crusher and passed through a 40 mesh sieve. Then the powdered birch was extracted with distilled water at 40 times the volume at normal temperature (25 °C) for 24 h and subsequently centrifuged at 4500 rpm for 10 min. Next, the supernatant was concentrated up to a certain volume in a rotary evaporator under reduced pressure. Finally, the supernatant was collected to obtain birch extract.

*Inonotus obliquus* was inoculated on potato dextrose agar (PDA) slants at 28 °C for 7–9 days. The mycelium obtained from slant pure culture was maintained at 4 °C and passaged every three months. The seed medium was composed of glucose 20 g/L, tryptone 4 g/L, KH_2_PO_4_ 1g/L and MgSO_4_ 1 g/L, shaking culture under 150 rpm at 28 °C. *I. obliquus* from the PDA slant were inoculated into a 250-mL flask containing 100 mL seed medium and cultured for 7 days. And the seed medium was inoculated into a 250-mL flask containing 100 mL fermentation medium and cultured for 13 days [27]. The compositions of fermentation medium were same as the seed medium and differing kinds and concentrations of stimulatory agents were evaluated in submerged fermentation of *I. obliquus*. 2, 3, 4, 5, and 6 μg/mL of VB_1_, VB_6_, betulin, and 10, 20, 40, 60, and 80 *v*/*v* of birch extract were added to the seed culture as fermentation medium, respectively, the final volume of the fermentation medium were 100 mL. 

### 4.3. Measurement of Mycelial Biomass

After 13 days the fermentation ended up, the mycelial biomass was separated from fermentation broth by filtration, washed with deionized water until colorless, dried in the oven at 60 °C and weighed to obtain the mycelial biomass [24].

### 4.4. Polysaccharide Content and Reducing Sugar Content in Fermentation Broth

Culture broth of *I. obliquus* under various culture conditions was collected on day 13 and concentrated up to a certain volume by rotary evaporator filtration. Then absolute ethanol was added (4:1, *v*/*v*) and store overnight. The crude polysaccharides were obtained after alcohol precipitation. The phenol-sulfuric acid method was used to determined polysaccharide content and 3,5-dinitrosalicylic acid colorimetry (DNS) was used to determine reducing sugar content in fermentation broth [28,29].

### 4.5. Assay of α-Glucosidase Inhibitory Activity

The α-glucosidase inhibitory activity of polysaccharide was determined as previously reported, with some modification [30,31]. First, 60 µL of phosphate buffer (pH 6.8) and 20 µL of 10, 20, 40, 80, 160, 320 μg/mL polysaccharide sample solution were mixed and incubated in a 96-well microtiter plate at 37 °C for 5 min. Then 20 µL of 2.5 mmol/L pNPG was added and incubated for another 30 min. The reaction was terminated by adding 30 μL of 0.1 mol/L Na_2_CO_3_ solution after adding 20 μL of 0.2 U/mL α-glucosidase and incubating at 37 °C for 30 min. The absorbance at 405 nm was measured using a multimode microplate reader. The inhibitory rates (%) were calculated according to the following formula, and the IC_50_ values were calculated with Origin 8.0:(1)$$R=\left(1-\frac{A3-A4}{A1-A2}\right)\times 100\%$$
where A1 was the absorbance of the blank, A2 was the absorbance of the blank control, A3 was the absorbance of the sample, A4 was the absorbance of the sample control.

### 4.6. Monosaccharide Composition Analysis

The monosaccharide composition was carried out following a reported method with some modification [32,33]. EPS (15 mg) was added with trifluoroacetic acid (2 mL 2 mol/L), hydrolyzed at 105 °C for 3 h. NaOH solution (500 μL 0.3 mol/L) and PMP solution (500 μL 0.5 mol/L) were added into 1 mL of the hydrolyzate, derivatized under 75 °C water bath at for 3 hours. Finally, the reaction system was neutralized with HCl solution (600 μL 0.3 mol/L), extracted with chloroform, repeated 5 times, then filtered the upper aqueous phase with 0.22 μm of filter membrane. The monosaccharide composition of *I. obliquus* EPS was determined by liquid chromatography. The standard stock solution consists of seven monosaccharide standards of glucose, rhamnose, galactose, arabinose, xylose, mannose, and fucose. The chromatographic conditions: C18 column (250 mm × 4.6 mm, 5 μm), phosphate buffer (20 mmol/L, pH6.7)—acetonitrile (82:18) as mobile phase with the flow rate of 1 mL/min, column temperature was 30 °C, and the detection wavelength was 250 nm. The monosaccharide content was calculated according to the calibration curve (peak area-concentration) of each monosaccharide standard.

### 4.7. Statistical Analysis

SPSS 22.0 software (IBM SPSS, Chicago, IL, USA) was used for the statistical analysis. The assays were carried out as three replicates, which from different biological experiments and analytical data was collected from each one of the biological experiments three times. *p* Values of less than 0.05 were considered statistically significant.

## 5. Conclusions

In this study, the effects of four stimulatory agents on mycelial growth and polysaccharide production in submerged fermentation of *I. obliquus* were investigated. The results showed that the addition of stimulatory agents could be used as a new regulatory method for polysaccharide biosynthesis in *I. obliquus*, which could not only promote the mycelium growth and EPS production, but also significantly increase the inhibitory activity of the resulting polysaccharides against α-glucosidase. Moreover, the addition of stimulatory agents had a significant effect on the monosaccharide composition of polysaccharides, whereby under the addition of different stimulatory agents, polysaccharides with different activity and different monosaccharide compositions were obtained. The addition of stimulatory agents in liquid submerged fermentation provides an experimental basis for industrial large-scale production of high-activity polysaccharides of *I. obliquus*, which is of great significance for studying its biological activity. 

In summary, the stimulating factors can affect hypoglycaemic activity of polysaccharides from *Inonotus obliquus* by submerged fermentation. The monosaccharide composition had a correlation with α-glucosidase inhibitory activity, indicating that the activity of polysaccharides was related to the structure of polysaccharides, but the synthesis of polysaccharides is a complex process. The initial carbon source in the medium provides sufficient substrate for the synthesis of polysaccharides of *Inonotus obliquus* and the addition of stimulating factors changed the composition of monosaccharides, which affect the activity of polysaccharides. Understanding the synthesis pathway of polysaccharides added by different stimulating factors is a long-term and complicated process, which can be guided by genetic engineering. In addition, for the structure of polysaccharides, only the monosaccharide composition has been studied in this paper, however, it is also necessary to explore the advanced structure of polysaccharides, using techniques such as nuclear magnetic resonance, FT-IR, molecular weight, the linkage mode of monosaccharides and glycosidic bond configuration in polysaccharides. Further studies to investigate the relationship between the structure and the α-glucosidase inhibitory activity are needed. 

## Figures and Tables

**Figure 1 molecules-24-04400-f001:**
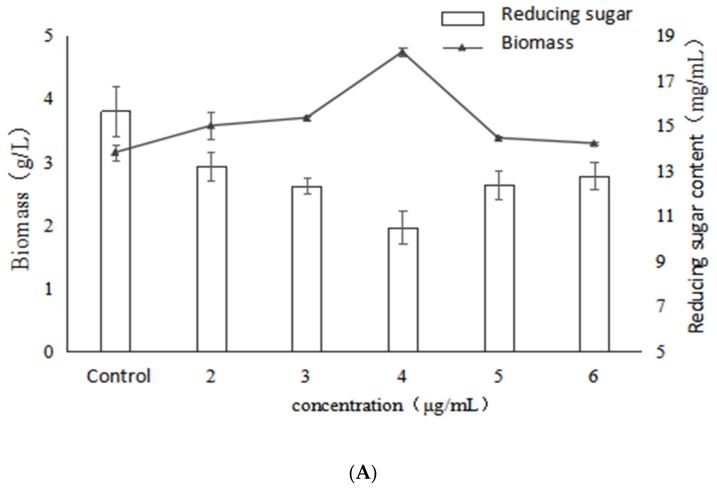
Effects of different concentrations of VB_6_ on growth, polysaccharide content and activity of *Inonotus obliquus* (**A**) biomass and reducing sugar content (**B**) polysaccharide content and α-glucosidase inhibition rate.

**Figure 2 molecules-24-04400-f002:**
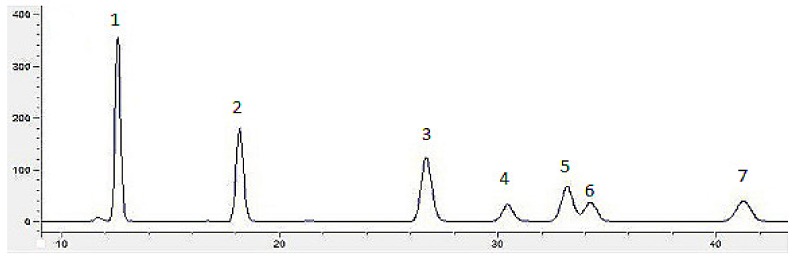
Chromatograms of monosaccharides compositions: seven monosaccharides (1: mannose, 2: rhamnose, 3: glucose, 4: galactose, 5: xylose, 6: arabinose, 7: fucose).

**Figure 3 molecules-24-04400-f003:**
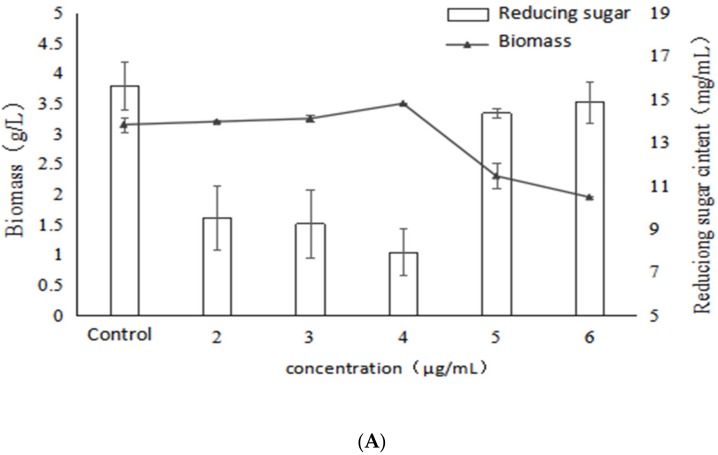
Effects of different concentrations of VB_1_ on growth, polysaccharide content and activity of *Inonotus obliquus* (**A**) biomass and reducing sugar content (**B**) polysaccharide content and α-glucosidase inhibition rate.

**Figure 4 molecules-24-04400-f004:**
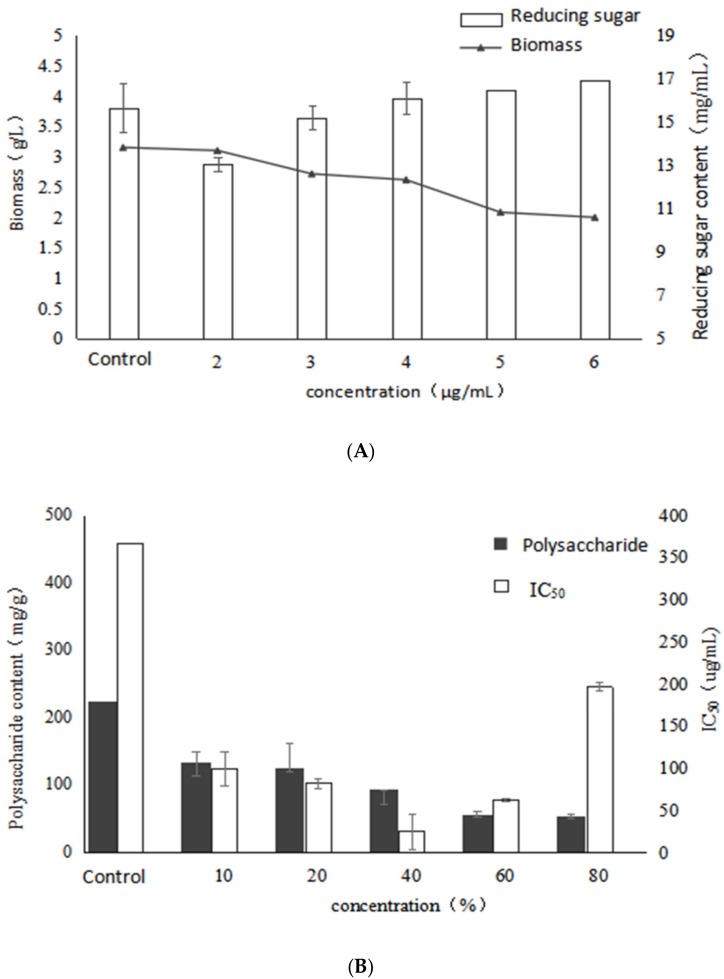
Effects of different concentrations of betulin on growth, polysaccharide content and activity of *Inonotus obliquus* (**A**) biomass and reducing sugar content (**B**) polysaccharide content and α-glucosidase inhibition rate.

**Figure 5 molecules-24-04400-f005:**
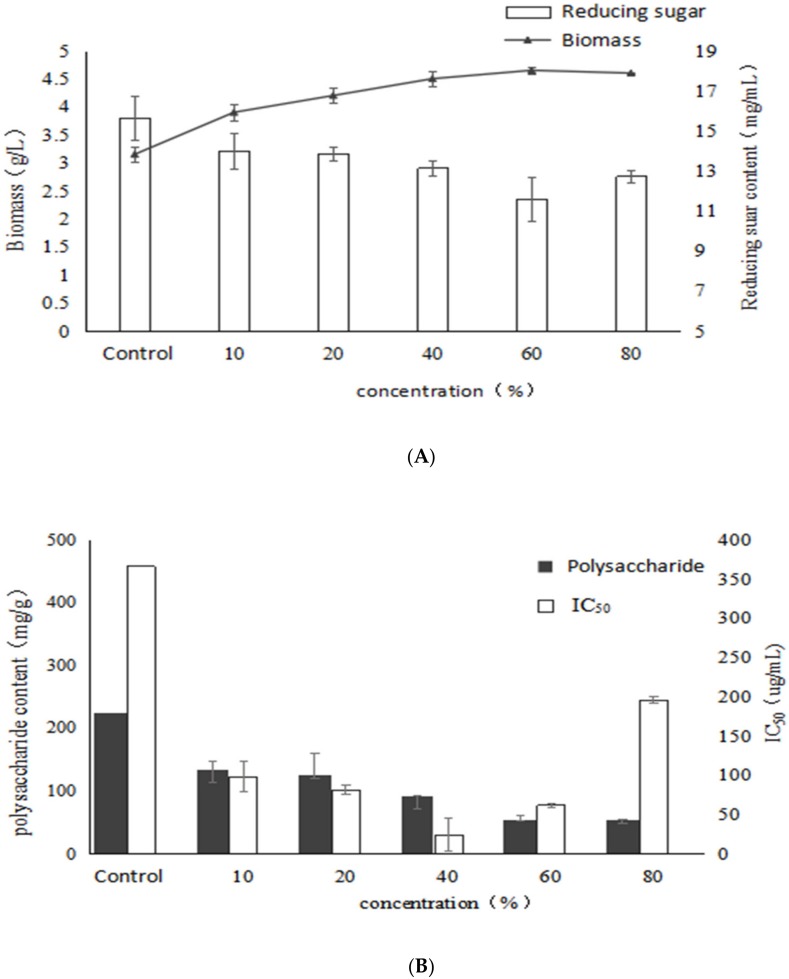
Effects of different concentrations of birch extract on growth, polysaccharide content and activity of *Inonotus obliquus* (**A**) biomass and reducing sugar content (**B**) polysaccharide content and α-glucosidase inhibition rate.

**Table 1 molecules-24-04400-t001:** Effects of Different Concentrations of VB_6_ on the composition of monosaccharides and molar ratio in *Inonotus obliquus.*

Concentrations (μg/mL)	2	3	4	5	6
Mannose (mol%)	18.83	18.40	21.66	26.79	28.63
Rhamnose (mol%)	19.45	23.94	27.93	26.25	24.26
Glucose (mol%)	33.84	25.01	15.14	12.94	12.57
Galactose (mol%)	-	-	-	-	-
Xylose (mol%)	-	-	-	-	-
Arabinose (mol%)	-	-	-	-	-
Fucose (mol%)	27.88	32.65	35.27	34.02	34.54

**Table 2 molecules-24-04400-t002:** Effects of Different Concentrations of VB_1_ on the composition of monosaccharides and molar ratio in *Inonotus obliquus.*

Concentrations (μg/mL)	2	3	4	5	6
Mannose (mol%)	13.18	20.76	19.84	19.13	18.61
Rhamnose (mol%)	17.07	27.13	17.07	19.26	20.84
Glucose (mol%)	25.01	26.08	31.97	32.44	32.68
Galactose (mol%)	19.61	-	-	-	-
Xylose (mol%)	-	-	-	-	-
Arabinose (mol%)	-	-	-	-	-
Fucose (mol%)	25.13	26.03	31.12	29.17	27.87

**Table 3 molecules-24-04400-t003:** Effects of Different Concentrations of betulin on the composition of monosaccharides and molar ratio in *Inonotus obliquus*.

Concentrations (μg/mL)	2	3	4	5	6
Mannose (mol%)	10.09	13.23	13.67	12.39	11.10
Rhamnose (mol%)	23.30	22.63	22.78	19.46	13.47
Glucose (mol%)	18.35	27.17	28.67	29.25	44.63
Galactose (mol%)	-	-	-	-	-
Xylose (mol%)	-	-	-	-	-
Arabinose (mol%)	14.50	15.71	13.05	10.98	15.34
Fucose (mol%)	33.76	21.26	21.84	26.92	15.46

**Table 4 molecules-24-04400-t004:** Effects of Different Concentrations of birch extract on the composition of monosaccharides and molar ratio in *Inonotus obliquus.*

Concentrations (%)	10	20	40	60	80
Mannose (mol%)	15.80	10.39	13.45	13.03	21.68
Rhamnose (mol%)	11.25	15.31	15.83	16.78	20.01
Glucose (mol%)	54.89	26.99	27.08	23.79	35.14
Galactose (mol%)	-	17.00	17.32	17.91	-
Xylose (mol%)	-	13.62	14.11	12.04	-
Arabinose (mol%)	18.07	15.99	12.21	16.44	23.09
Fucose (mol%)	-	-	-	-	-

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
