# Peer review of "Simultaneous Use of Stimulatory Agents to Enhance the Production and Hypoglycaemic Activity of Polysaccharides from Inonotus obliquus by Submerged Fermentation"

_molecules, 2019, doi:10.3390/molecules24234400_

Round 1

Reviewer 1 Report

This research on " Use of Stimulatory Agents Simultaneously To Enhance the Production and Hypoglycaemic Activity of Polysaccharides from Inonotus obliquus by Submerged Fermentation" investigates the effect of some stimulatory agents on the accumulation of exo-polysaccharides and their monosaccharide composition by submerged fermentation of I. obliquus.

It presents interesting results and the contents are located in the field and scope of this journal.

The following are my minor comments:

Line 33: Author of species name [Inonotus obliquus (Fr.) Pilát] must be provided when the scientific name of the fungal species is first mentioned (see http://www.speciesfungorum.org/Names/SynSpecies.asp?RecordID=315905).

Line 159: leave a space between 6 and µg.

Line 211 (Table 3): correct formatting

Line 240: replace ‘brich’ with ‘birch’.

Line 245: the name of the Inonotus obliquus must be written in italics.

Line 254: leave a space before ‘Zeng[22]’.

Line 417: delete ‘29’.

Line 420: delete ‘30’.

Line 423: replace ‘S.G. Jonathan; I.O. Fasidi’ with ‘Jonathan, S.G.; Fasidi, I.O.’.

Line 424: replace comma with point (….. mushroom,Food Chemistry).

Line 435: leave a space before ‘Food Chem 2011’.

Line 443: leave a space before ‘Int J Biol Macromol’.

Author Response

Dear Reviewer:

On behalf of my co-authors, we appreciate you very much for their positive and constructive comments and suggestions on our manuscript entitled “Use of Stimulatory Agents Simultaneously To Enhance the Production and Hypoglycaemic Activity of Polysaccharides from Inonotus obliquus by Submerged Fermentation”

Line 33: Author of species name [Inonotus obliquus (Fr.) Pilát] must be provided when the scientific name of the fungal species is first mentioned (see http://www.speciesfungorum.org/Names/SynSpecies.asp?RecordID=315905).

 Inonotus obliquus (Fr.) Pilát is an edible fungus with medicinal properties.

Line 159: leave a space between 6 and µg.

A space has been left between 6 and ug.

Line 211 (Table 3): correct formatting

 The formatting of table 3 has been corrected.

Line 240: replace ‘brich’ with ‘birch’.

  ‘brich’ has been replaced by ‘birch’

Line 245: the name of the Inonotus obliquus must be written in italics.

 The name of the Inonotus obliquus has been rewritten in italics.

Line 254: leave a space before ‘Zeng[22]’.

 A space has been left before ‘Zeng[22]’.

Line 417: delete ‘29’.

 ‘29’ has been deleted.

Line 420: delete ‘30’.

  ‘30’ has been deleted.

Line 423: replace ‘S.G. Jonathan; I.O. Fasidi’ with ‘Jonathan, S.G.; Fasidi, I.O.’

  ‘S.G. Jonathan; I.O. Fasidi’ has been replaced by‘Jonathan, S.G.; Fasidi, I.O.’

Line 424: replace comma with point (….. mushroom,Food Chemistry).

  Comma has been replaced by point.

Line 435: leave a space before ‘Food Chem 2011’.

 A space has been left before ‘Food Chem 2011’.

Line 443: leave a space before ‘Int J Biol Macromol’.

 A space has been left before ‘Int J Biol Macromol’.

Reviewer 2 Report

This paper deals with the study of adding some vitamins (B1, B6) or betulin or birch extract, to enhance the mycelium and EPS production by Inonotus obliquus. The impact on monosaccharide composition of EPS is also highlighted.

As a general comment, I consider the paper would need major revision before being suitable for publication. However, the topic is interesting.

-The paper needs to be checked by native-English speaker, as some sentences are difficult to understand.

-Main comment will be that in my opinion results are not often discussed and that some conclusions should be removed or modified. As an example, authors claim that the higher rhamnose content in EPS produced using VB1 is responsible of an enhanced activity (lines 119-120), while it as no impact on activity of EPS produced using VB6 (lines 166-167). These conclusions are contradictory as if the biological activity is linked to the presence of a specific monosaccharide, presence of rhamnose should be correlated to the activity in both cases. Authors should discuss their results regarding other studies for which a specific monosaccharide in an EPS composition have been shown to modulate a biological activity. Moreover, mechanism by which the tested molecules and extract can have an impact on growth, EPS synthesis, and EPS composition, would be of interest and hypothesis should be proposed.

-The EPS should be more characterized in their composition (uronic acids, sulphate, proteins?), as presence of such compounds could have an impact on biological activity.

Some other comments to improve the manuscript:

-Mat and Met:

            -section 4.2: what is the difference between “seed medium” and “fermentation medium”? only the stimulatory agent adding?

            -section 4.2: the birch extract: please clarify the preparation method of the birch extract (grinding? Then granulometry of particles? Water extraction but at which temperature? Under mixing? Filtration with which cut-off?)

            -section 4.5, line 311: “20µL of polysaccharide sample”: at which concentration? Is the same concentration for all tests?

            -section 4.5, line 314: “20µL of 0.2U/mL and incubating” please add “alpha glucosidase” after 0.2U/mL

            -section 4.6: please add details about standards used and standard curves (which monosaccharides and concentrations)

Author Response

Dear Reviewer:

On behalf of my co-authors, we appreciate you very much for their positive and constructive comments and suggestions on our manuscript entitled “Use of Stimulatory Agents Simultaneously To Enhance the Production and Hypoglycaemic Activity of Polysaccharides from Inonotus obliquus by Submerged Fermentation”

Authors should discuss their results regarding other studies for which a specific monosaccharide in an EPS composition have been shown to modulate a biological activity.

 Kim [14] believe that rhamnose have a greater impact on the biological activity of polysaccharides from Inonotus obliquus.Furthermore, the change of monosaccharide composition of fermented polysaccharide may be one of the reasons for the change of polysaccharide activity [21].

Moreover, mechanism by which the tested molecules and extract can have an impact on growth, EPS synthesis, and EPS composition, would be of interest and hypothesis should be proposed.

Studying the differences in gene expression profiles and metabolic differences of liquid fermentation of Inonotus obliquus under the condition of adding stimulating factors can clarify the molecular mechanism of regulation and metabolism of Inonotus obliquus. The mechanism may lay a theoretical foundation for the realization of metabolic regulation at the gene level.

As an example, authors claim that the higher rhamnose content in EPS produced using VB1 is responsible of an enhanced activity (lines 119-120), while it as no impact on activity of EPS produced using VB6 (lines 166-167).

These conclusions are contradictory as if the biological activity is linked to the presence of a specific monosaccharide, presence of rhamnose should be correlated to the activity in both cases.

results are not often discussed and that some conclusions should be removed or modified.

Results have been rewritten and conclusions have been modified.

-The EPS should be more characterized in their composition (uronic acids, sulphate, proteins?), as presence of such compounds could have an impact on biological activity.

uronic acids and proteins of Inonotus obliquus have been studied in our previous study. ‘Chemical characterization and hypoglycaemic activities in vitro of two polysaccharides from Inonotus obliquus by submerged culture. Molecules 2018, 23,3261’

 -section 4.2: what is the difference between “seed medium” and “fermentation medium”? only the stimulatory agent adding?

The compositions of fermentation medium were same as the seed medium and differing kinds and concentrations of stimulatory agents were evaluated in submerged fermentation of I.obliquus. 2, 3, 4, 5, and 6 μg/mL of VB1, VB6, betulin, and 10, 20, 40, 60, and 80 vol/vol of birch extract were added to the seed culture as fermentation medium, respectively, the final volume of the fermentation medium were 100 ml .

 -section 4.2: the birch extract: please clarify the preparation method of the birch extract (grinding? Then granulometry of particles? Water extraction but at which temperature? Under mixing? Filtration with which cut-off?)

After drying in an oven at 60 °C, the birch trunk was broken with a crusher and passed through a 40 mesh sieve. Then the powdered birch was extracted with distilled water at 40 times the volume at normal temperature (25°C) for 24 h and subsequently centrifuged at 4500 rpm for 10 min. Next, the supernatant was concentrated up to a certain volume in a rotary evaporator under reduced pressure. Finally, the supernatant was collected to obtain birch extract.

 -section 4.5, line 311: “20µL of polysaccharide sample”: at which concentration? Is the same concentration for all tests?

20 µL of 10, 20, 40, 80, 160, 320 μg/mL polysaccharide sample solution were mixed and incubated in a 96-well microtiter plate at 37 °C for 5 min.

 -section 4.5, line 314: “20µL of 0.2U/mL and incubating” please add “alpha glucosidase” after 0.2U/mL

The reaction was terminated by adding 30 μL of 0.1 mol/L Na2CO3 solution after adding 20 μL of 0.2 U/mL α-glucosidase and incubating at 37 °C for 30 min.

  -section 4.6: please add details about standards used and standard curves (which monosaccharides and concentrations)

 The standard stock solution consists of seven monosaccharide standards of glucose, rhamnose, galactose, arabinose, xylose, mannose, and fucose. The monosaccharide content was calculated according to the calibration curve (peak area-concentration) of each monosaccharide standard.

Reviewer 3 Report

Authors have studied the influence of several "stimulatory agents" on mycelial biomass as well as production and composition of crude polysaccharide mixtures by a fungus. The polysaccharide mixtures were investigated for alpha-glucosidase inhibition. Authors claim that the addition of vitami B1 and B6, as well as betulin, and "birch extract" alter the monosaccharide composition of exopolysaccharides and alpha-glucosidase activity.

Originality:

Similar work has been published by Xiang-qun Xu et al. (see the references)

General comments:

The work is incomplete. The paper is descriptive and speculative and does not contain molecular aspects. The results are summarized again in the conclusions section which, otherwise, lacks information on the molecular mechanism of the claimed polysaccharide synthesis regulation.

The rationale behind using vitamin B, betulin, and birch extract  must be explained. Is there precedence in the literature ? It also must be explained in the conclusions, why these "stimulants" alter the composition of crude polysaccharides and how this could be effected on a molecular level. The statement that the structure and mechanism of action needs further investigation is obvious but insuficcient.

The English needs much improvement.

The paper is not acceptable for publication in "Molecules" in its present form. It must be completely rewritten in a sound scientific style, in case authors wish to resubmit a revised version. Please have it checked by a native English speaking chemist.

Specific comments:

All Tables: Indicate the unit: mol% or weigt % ?. Monosaccharide composition is missing for the controls. Inonotus polysaccharides of have been described extensively in the literature. It is difficult to understand why the polysaccharides investigated here were not fractionated and purified before analysis of monosaccharide composition.

Line 123: "one of the reasons that affected polysaccharide activity may be a simulating factor that altered monosaccharide composition[21]." What is a simulating factor ? And how would it affect polysaccharide activity ?

Line 191: "The addition of betulin caused arabinose in the monosaccharide composition of polysaccharides." Impossible sentence ! It is not logical and wrong as such.

Line 276: "multiple UDP-monosaccharides are combined into polysaccharides under the action of glycosidic bonds." This does not make sense. What do you mean with "action of glycosidic bonds" ?

Experimental:

The birch extract was obtained by water extraction for 8-12 h and filtration: Which part of the plant was extracted ? How is the percentage of "birch extract" defined: vol/vol ? or gramm per vol ? When 80% was added, what is the final volume ? Was the extract evaporated and dried, or was it used without further processing. Please describe exactly what has been done.

Also specify what is the composition of the extract. Are there mono- and polysaccharides in the extract ? What is the monosaccharide composition of the birch extract and how would this influence the results ?

Statistics: are the replicates from different biological experiments or from analytical data collection of one biological experiment ?

Some of the typing errors:

Line 145:  concertration must read concentration

Line 244: extracton must read extract on

Author Response

Dear Reviewer:

On behalf of my co-authors, we appreciate you very much for their positive and constructive comments and suggestions on our manuscript entitled “Use of Stimulatory Agents Simultaneously To Enhance the Production and Hypoglycaemic Activity of Polysaccharides from Inonotus obliquus by Submerged Fermentation”

Originality:

Similar work has been published by Xiang-qun Xu et al. (see the references)

Xiang-qun Xu et al investigated the effect of fatty acids, plant oils, organic solvents, and surfactants to improve the antioxidant activity of polyphenols from Inonotus obliquus,which are different from our study.

[Xu, X.; Shen, M.; Quan, L. Stimulatory Agents Simultaneously Improving the Production and Antioxidant Activity of Polyphenols from Inonotus obliquus by Submerged Fermentation. Appl Biochem Biotech 2015, 176, 1237-1250.]

General comments:

The work is incomplete. The paper is descriptive and speculative and does not contain molecular aspects. The results are summarized again in the conclusions section which, otherwise, lacks information on the molecular mechanism of the claimed polysaccharide synthesis regulation.

The rationale behind using vitamin B, betulin, and birch extract  must be explained. Is there precedence in the literature ?

Yes, there are some studies focus on the vitamin B, betulin, and birch extract. Please see the introduction.

 It also must be explained in the conclusions, why these "stimulants" alter the composition of crude polysaccharides and how this could be effected on a molecular level. The statement that the structure and mechanism of action needs further investigation is obvious but insuficcient.

In summary, the stimulating factors can affect hypoglycaemic activity of polysaccharides from Inonotus obliquus by submerged fermentation. The monosaccharide composition had a correlation against α-glucosidase inhibitory activity, indicating that the activity of polysaccharide related to the structure of polysaccharides, but the synthesis of polysaccharides is a complex process. The initial carbon source in the medium provides sufficient substrate for the synthesis of polysaccharides of Inonotus obliquus and the addition of stimulating factors promotes the catalysis of different enzymes to affect the composition of monosaccharides. The composition of monosaccharides affects the activity of polysaccharides. Understanding the synthesis pathway of polysaccharides added by different stimulating factors is a long-term and complicated process, which can be guided by genetic engineering. In addition, for the structure of polysaccharides, only the monosaccharide composition had been studied in this paper, however, it was also necessary to explore the advanced structure of polysaccharides, such as nuclear magnetic resonance, FT-IR, molecular weight, the linkage mode of monosaccharides and glycosidic bond configuration in polysaccharides. Further studies to investigate the relationship between the structure and the α-glucosidase inhibitory activity are wanted.

Specific comments:

All Tables: Indicate the unit: mol% or weigt % ?.

All Tables have been indicated the unit.

Monosaccharide composition is missing for the controls.

This paper mainly studies the comparison of monosaccharide composition under different concentrations of stimulating factors,therefore, the monosaccharide composition of the controls were not present.

 Inonotus polysaccharides of have been described extensively in the literature. It is difficult to understand why the polysaccharides investigated here were not fractionated and purified before analysis of monosaccharide composition.

In this paper, we focus on the hypoglycemic activity and monosaccharide composition of crude polysaccharides under the action of stimulating factors. Therefore, we first study the monosaccharide composition of crude polysaccharides, and we will further study on purified polysaccharides and there structure later.

Line 123: "one of the reasons that affected polysaccharide activity may be a simulating factor that altered monosaccharide composition[21]." What is a simulating factor ? And how would it affect polysaccharide activity ?

 It indicated that there is a certain relationship between monosaccharide composition and polysaccharide activity. Furthermore, the change of monosaccharide composition of fermented polysaccharide may be one of the reasons for the change of polysaccharide activity.

Line 191: "The addition of betulin caused arabinose in the monosaccharide composition of polysaccharides." Impossible sentence ! It is not logical and wrong as such.

 It can be seen from the table 3 that when adding betulin, a certain proportion of arabinose appears in the monosaccharide composition of the polysaccharides of I. obliquus.

Line 276: "multiple UDP-monosaccharides are combined into polysaccharides under the action of glycosidic bonds." This does not make sense. What do you mean with "action of glycosidic bonds" ?

The synthesis of polysaccharides is a complex process. The initial carbon source in the medium provides sufficient substrate for the synthesis of polysaccharides from Inonotus obliquus. Under the catalysis of enzymes, glucose acts as a sugar donor, while multiple UDP-monosaccharides form polysaccharides under the action of glycosidic bonds.The addition of stimulating factors promote the catalysis of different enzymes to affect the composition of monosaccharides, which affect the activity of polysaccharides.

Experimental:

The birch extract was obtained by water extraction for 8-12 h and filtration:

Which part of the plant was extracted ?

 Birch bark was purchased from Birch forest factory of Changbai Mountain, Jilin Province.

 How is the percentage of "birch extract" defined: vol/vol ? or gramm per vol ?

When 80% was added, what is the final volume ?

10, 20, 40, 60, and 80 vol/vol of birch extract were added to the seed culture as fermentation medium, respectively, the final volume of the fermentation medium were 100 ml .

Was the extract evaporated and dried, or was it used without further processing. Please describe exactly what has been done.

After drying in an oven at 60 °C, the birch bark was broken with a crusher and passed through a 40 mesh sieve. Then the powdered birch was extracted with distilled water at 40 times the volume at normal temperature (25°C) for 24 h and subsequently centrifuged at 4500 rpm for 10 min. Next, the supernatant was concentrated up to a certain volume in a rotary evaporator under reduced pressure. Finally, the supernatant was collected to obtain birch extract.

Also specify what is the composition of the extract. Are there mono- and polysaccharides in the extract ? What is the monosaccharide composition of the birch extract and how would this influence the results ?

The nutrients of birch extract were studied by HPLC, and it was found that nearly 70 kinds of compounds contained in birch extract mainly contained various vitamins, amino acids, fatty acids and mineral elements.

[54]Kallio, H. Aroma of birch syrup. J Agr Food Chem 1989, 5, 1367-1371

Statistics: are the replicates from different biological experiments or from analytical data collection of one biological experiment ?

The assays were carried out as three replicates, which from different biological experiments and the analytical data was collected from each one of the biological experiment three times. 

Some of the typing errors:

Line 145:  concertration must read concentration

Concertration has been replaced by concentration

Line 244: extracton must read extract on

 Extracton has been replaced by extract on.

Reviewer 4 Report

This work describes use of stimulatory agents simultaneously to enhance the production and hypoglycaemic activity of polysaccharides from Inonotus obliquu by submerged fermentation.

Generally, the article its quite well written but needs some improvement.

- In the article lacks information on birch extract, where it was taken from and what its composition was. After all, this extract also contains sugars that culd have an impact on the parameters studied as polysaccharide content, reducing sugar content and α-glucosidase inhibition rate.

- In the sentence on the page 14, line 191, he verb is missing.

- Spaces must be checked throughout the article because they are often missing, especially before quotation brackets.

Author Response

Dear Reviewer:

On behalf of my co-authors, we appreciate you very much for their positive and constructive comments and suggestions on our manuscript entitled “Use of Stimulatory Agents Simultaneously To Enhance the Production and Hypoglycaemic Activity of Polysaccharides from Inonotus obliquus by Submerged Fermentation”

- In the article lacks information on birch extract, where it was taken from and what its composition was. After all, this extract also contains sugars that could have an impact on the parameters studied as polysaccharide content, reducing sugar content and α-glucosidase inhibition rate.

Birch bark was purchased from Birch forest factory of Changbai Mountain, Jilin Province. After drying in an oven at 60 °C, the birch trunk was broken with a crusher and passed through a 40 mesh sieve. Then the powdered birch was extracted with distilled water at 40 times the volume at normal temperature (25°C) for 24 h and subsequently centrifuged at 4500 rpm for 10 min. Next, the supernatant was concentrated up to a certain volume in a rotary evaporator under reduced pressure. Finally, the supernatant was collected to obtain birch extract.

The nutrients of birch extract were studied by HPLC, and it was found that nearly 70 kinds of compounds contained in birch extract mainly contained various vitamins, amino acids, fatty acids and mineral elements.

[Kallio, H. Aroma of birch syrup. J Agr Food Chem 1989, 5, 1367-1371]

In the sentence on the page 14, line 191, he verb is missing.

 he verb was added on the page 14, line 19.

 Spaces must be checked throughout the article because they are often missing, especially before quotation brackets.

Spaces have been checked throughout the article.

Round 2

Reviewer 2 Report

The manuscript has been significantly improved. I still consider that some parts should have been more discussed but it is now suitable for publication

Author Response

Dear Reviewer:

On behalf of my co-authors, we appreciate you very much for their positive and constructive comments and suggestions on our manuscript entitled “Use of Stimulatory Agents Simultaneously To Enhance the Production and Hypoglycaemic Activity of Polysaccharides from Inonotus obliquus by Submerged Fermentation”

We have made a few changes on this paper, please see the manuscript.

Reviewer 3 Report

Please find comments in the attached "Reviewer Comments to Revised Version V2.PDF".

Authors may condider to submit to a journal specialized in food and nutrition.
